# Estimation of Health Effects and Economic Losses from Ambient Air Pollution in Undeveloped Areas: Evidence from Guangxi, China

**DOI:** 10.3390/ijerph16152707

**Published:** 2019-07-29

**Authors:** Feng Han, Xingcheng Lu, Cuicui Xiao, Miao Chang, Ke Huang

**Affiliations:** 1Key Laboratory of Beibu Gulf Environment Change and Resources Utilization of Ministry of Education, Nanning Normal University, Nanning 530001, China; 2Division of Environment, Hong Kong University of Science & Technology, Clear Water Bay, Hong Kong, China; 3School of Environmental, Tsinghua University, Beijing 100084, China; 4Guangxi Environmental Protection Bureau, No.16 of Foziling Road, Nanning 530001, China

**Keywords:** air pollution, health damage, economic loss, underdeveloped region, urbanization and industrialization

## Abstract

Guangxi Zhuang Autonomous Region, located in the southwest of China, has rapidly developed since the late 2000s. Similar to other regions, economic development has been accompanied by environmental problems, especially air pollution, which can adversely affect the health of residents in the area. In this study, we estimated the negative health effects of three major ambient pollutants, Particulate Matter with a diameter of 10 μm or less (PM10), Sulfur Dioxide (SO_2_) and Nitrogen Dioxide (NO_2_) in Guangxi from 2011 to 2016 using a log-linear exposure–response function. We monetarized the economic loss using the value of statistical life (VSL) and the cost of illness (COI) methods. The results show that the total possible short-term all-cause mortality values due to PM10, SO_2_, and NO_2_ were 28,396, with the confidence intervals from 14,664 to 42,014 (14,664–42,014), 24,618 (15,480–33,371), and 46,365 (31,158–61,423), respectively. The mortality from the three pollutants was 48,098 (19,972–75,973). The economic loss of the health burden from the three pollutants was 40,555 (24,172–57,585), which was 2.86% (1.70–4.06%) of the regional gross domestic product. The result of the comparative analysis among different cities showed that urbanization, industrialization, and residents’ income are important factors in air-pollution-caused health damage and subsequent economic loss. We conclude that the health burden caused by ambient pollutants in developing regions, accompanied by its rapid socio-economic growth, is significant and tighter regulation is needed in the future to alleviate air pollution and mitigate the related health damage.

## 1. Introduction

Since the implementation of the national strategy for advancing the development of Guangxi’s Beibu Gulf along and the western development strategy, the Guangxi Zhuang Autonomous Region (ZAR) in Southwest China (Figure 1) has been enjoying considerable economic growth. The Guangxi region has a host of unique geographic advantages: it is adjacent to Guangdong, one of the most affluent provinces in China; it is contiguous with other Association of Southeast Asian Nation (ASEAN) countries, and it incorporates Western China’s distinctive estuary. Consequently, it is becoming a significant bridge between China and the ASEAN nations as well as a key urban agglomeration in Western China. In 2016, the gross domestic product (GDP) of Guangxi ZAR reached 1.83 trillion RMB (USD $267.7 billion) [1], similar to developing countries such as Chile.

Inevitably, economic surges take a toll on the environment. From 1998 to 2012, environmental pollution in Guangxi considerably worsened, the emission of industrial waste gas increased nearly six times from 415.2 billion to 2761 billion m^3^ [2]. In 2013, the air pollution index of Guangxi’s capital, Nanning, showed traces of fine particulates (PM2.5), indicating high levels of air pollution [3]. Other major cities in Guangxi, such as Yulin, Liuzhou and Guilin, also exhibit deteriorating air quality conditions [2]. A monitoring study [4] found that the concentration of NO_2_, carbon monoxide (CO), and hydrocarbons in some districts of Nanning exceeded the national standard by 8 to 15 times, and the pollutants were accumulating on the leeward side of buildings due to the wind, endangering human health. According to the official monitoring data by Guangxi environmental protection department, the over 50% daily average concentration of PM10 in the 14 cities of Guangxi acceded 50 μg/m^3^ (national first grade standard), and almost 5% daily average concentration of PM10 acceded 150 μg/m^3^ (national second grade standard), across 2015 [5].

Many studies have clarified the impact of air pollution on human health. Epidemiological studies have shown that air pollutants, such as SO_2_, NO_2_, PM10, and others, can damage the human respiratory and cardiovascular systems and contribute to premature death [6]. Cohen, et al. [7] summarized that PM2.5 is responsible for about 3% of the global mortality from cardiopulmonary diseases; about 5% of cancers of the trachea, bronchus, and lung; and around 1% of acute respiratory infections in children under the age of five. In addition, according to Cohen, et al., the largest concentrations of PM2.5 were predominantly found in developing countries: 65% of total global PM2.5 was from Asia alone. Numerous studies have confirmed the negative effect of air pollutants (NO_2_, SO_2_, O_3_, and others) on respiratory and cardiovascular deaths and on hospital admissions for related diseases [8,9]. The economic costs of labor resource loss and medical expenses from these adverse health effects must also be considered.

Several studies have attempted to quantify the adverse health effects of ambient pollutants. Cao, et al. [10] conducted a time series analysis to examine the association of outdoor air pollutants (PM10, SO_2_, and NO_2_) with hospital outpatient and emergency room visits in Shanghai and discovered that outdoor air pollution (SO_2_ and NO_2_) is associated with an increase in hospital outpatient and emergency room visits. The time-series study conducted by Chen, et al. [11] showed that short-term exposure to outdoor air pollution is associated with cardiovascular-related hospital admissions, and they determined the coefficients accordingly. Kan, et al. [12] suggested that ambient air pollutants (PM10, SO_2_, and NO_2_) have both short- and long-term effects on cardiovascular disease in China, asserting that the cardiovascular risks in China are more significant than that in North America or Europe.

The range of health effects from air pollutants includes premature mortality, increased hospital admissions, and increased outpatient visits, all of which negatively impact economic productivity. They result in a reduction of total working hours in the labor force and an increase in medical expenses, which detract from economic and social welfare. According to Matus, et al. [13], air pollution in China has had a substantial burden on the economy, with ozone (O_3_) and PM10 concentrations levels alone leading to USD $16–69 billion (or 7–23%) loss of consumption and USD $22–112 billion (or 5–14%) loss of welfare in China’s economy. Xia, et al. [14] established a supply driven input-output (I-O) model to estimate total output losses’ monetary value resulting from air pollution disease-related reductions in working hours across 30 Chinese provinces in 2007. They discovered a total economic loss of 346.26 billion Yuan (approximately 1.1% of the national GDP) from PM2.5 in 2007 based on the number of affected Chinese employees (72 million out of a total labor population of 712 million), whose years of labor were reduced due to mortality, hospital admissions, and outpatient visits. Huang, et al. [15] applied COI and amended human capital (AHC) models to evaluate the health risk of PM10 in the Pearl River Delta (PRD). They concluded that in 2006, the total economic loss from PM10 pollution-related health effects in the PRD region was 29.21 billion Yuan, equivalent to 0.72% of the regional GDP, with premature death and chronic respiratory disease accounting for over 95% of the total loss. Lu, et al. [16] estimated that the total economic loss from the negative health effects caused by four pollutants (SO_2_, NO_2_, O_3_, and PM10) ranged from USD $14,768 to 25,305 million, accounting for 1.4–2.3% of the PRD GDP. Zhao, et al. [17] estimated the 2014 economic loss from PM10 in Beijing and showed that a loss of USD $583.02 million (or 0.03% of its GDP) resulted from depreciation in human capital due to premature death.

To date in China, studies on the negative health effects of air pollutants and their corresponding economic losses have focused either on the whole country or on relatively developed regions (e.g., the PRD region) [15,16,18]. However, research on the southwestern area, including the Guangxi province, remains scarce. Compared to developed regions in China, such PRD region, Shanghai, Beijing, e.g., where GDP per capita has reached 20,000 US dollars in 2016 [19], the GDP per capita of Guangxi in 2016 was only 5522 US dollars [1], approximately two-thirds of the average level in China, and half of the global average [20]. As underdeveloped area, Guangxi is under pressure from both economic development and environmental protection. For the lack of adequate medical service to part of the population in Guangxi, their health might more sensitive to air pollution.

Thus, our study attempted to estimate the health damage and the related economic loss, including loss of labor resource and cost of medical treatment, due to air pollution in Guangxi ZAR. This article can help to answer the following questions: (1) What is the magnitude of estimated health burden, including morbidity, mortality and economic loss in the underdeveloped area like Guangxi, where was scarcely considered by previous studies? (2) What pollutant contributed to the greatest health damage, among the three ambient pollutants (PM10, SO_2_, NO_2_), in Guangxi from 2011 to 2016? (3) How is the health burden varied over the 14 cities in Guangxi while big gaps in socio-economic development exist among them? and (4) what effect of socio-economic factors contribute to the health burden caused by ambient pollution?

## 2. Materials and Methods

### 2.1. Estimating the Health Effect

The exposure-response (E-R) function, which is widely used for health damage assessment, has been applied to calculate the health effects of ambient pollutant [15,16,21]. In this study, we only estimated the short-term health effects of ambient pollutants; long-term effects were not included. The endpoint of short-term health damage was categorized into premature mortality, hospital admissions, and outpatient visits. Premature mortality included cardiovascular disease mortality (mortality for CVD) and respiratory mortality (mortality for RD); hospital admission included hospital admission for cardiovascular disease (inpatients for CVD) and hospital admission for respiratory disease (inpatients for RD) as subclasses. The morbidity and mortality due to CVD and RD were emphasized in this article because they are the disease with high mortality rate and the causality between these two diseases and air pollution has been extensively tested and verified by epidemiological studies [6,7,22,23], thus they are typical diseases characterizing the health effects of air pollution.

In this study, the E–R function is expressed by the equation:ΔY = Y_0_ × (e^βΔX^ − 1) × Pop(1)
where ΔY represents health effects, such as non-accident mortality or hospital admission due to related ambient air pollutants; Y_0_ is the baseline incidence rate of specific health problem per unit population taken from the statistical yearbook of the National Health and Planning Commission [24]; and β is the exposure–response coefficient of ambient pollution from the literature, indicating an increase in cases of adverse health effects per 10 μg/m^3^ increase in ambient air pollutant (Table 1). The coefficient selection was the core of this study as it greatly affects the magnitude of the eventual valuation results. Since no coefficient for the Guangxi region could be found in the current literature, we applied a coefficient applicable to the western region or the whole country (Table 1).

Δx in Equation (1) represents the difference between the true exposure concentration and the concentration of ambient pollution that causes no adverse health effects, also known as the safe concentration. Since definitions of safe thresholds for ambient pollutants are still controversial, we adopted lower limits as the natural background concentrations of related air pollutants (shown in Appendix A) [25]. Pop represents the population of the study area, which was the resident population of 14 municipal cities in Guangxi, as shown in Appendix A. Concentrations of the three ambient air pollutants were obtained from the monitoring data provided by Guangxi provincial environmental departments. We changed the daily average concentrations of air pollutants into monthly and annual average concentrations to simplify the research process. The example data are provided in Appendix A.

### 2.2. Monetarizing the Health Damage

We adopted VSL and COI methods to evaluate the economic loss due to health damage caused by ambient pollutants. The VSL method was applied to assign economic value to a certain person. As we know, life and health cannot be weighed monetarily, VSL is an economic value measuring the breakeven point that society, as a whole, is willing to pay for reducing the statistical risk of death [29]. According to a report by OECD [30], the VSL for China was 0.98 million US dollars in 2010. However, Wang and He concluded that estimated VSL for China in 2010 was only USD $0.13 million [29]. This article adopted the VSL of Wang’s research since they conducted a household survey on public health and environment in low-income areas in China which made the results more in line with the actual situation of China’s underdeveloped areas. Therefore, according to Wang’s estimation, the VSLs from 2011 to 2016 were calculated based on the annual Consumer Price Index (CPI), shown in Appendix A. The VSLs from 2011 to 2016 was USD $0.14 million, 0.16 million, 0.161 million, 0.169 million, 0.17 million, and 0.162 million, respectively.

COI method directly values the minimum economic burden of health damage by calculating the various disease-related expense, including the cost of pharmaceutical, outpatient service and hospitalization, and plus the loss of income due to sick leave. Cost of illness (COI) method is widely used to estimate the cost of different diseases in various regions [15,17,21,29].

The COI for all-cause hospital admissions and outpatient were calculated using Equation (2):C = (C_m_ + GDP_d_ × T) × ΔY(2)
where C is the total economic loss of hospital admissions caused by exposure to ambient air pollutants, C_m_ is the cost for medical treatment for each case, GDP_d_ is the daily GDP per capita, T refers to labor time loss due to illness, and ΔY represents the number of cases of hospital admission. The C_m_ and T (inpatient days) from 2010 to 2016, cited from the statistical yearbooks published by the National Health and Family Planning Commission, are shown in Appendix A.

The pollution and residential income information was acquired from the Guangxi provincial statistical yearbooks. The natural incidence rate of the endpoint of health impact (all-cause mortality, mortality for CVD, mortality for RD, hospital admission, outpatient visits, etc.), the statistical data of medical expenses, and treatment day of the hospital admission were quoted and derived from the China Health Statistics Yearbook.

In addition to analyzing the health effects of air pollutants, we also estimated the total health damage caused by the three pollutants. The different health effects from a single pollutant could not be directly added into the total health effect to avoid an overlapping issue, in the sense that a negative health effect may have been associated with more than one pollutant. Rather, health effects that were caused by multiple pollutants needed to be considered. A potential solution for the overlapping issue involved importing the exposure–response coefficients derived from the multi-pollutant model. However, up to now, few previous studies in China have applied three pollutant models. Instead, they usually applied two major pollutant models (NO_2_, PM10, PM2.5, SO_2_, O_3_, and coupled models) to calculate the associated coefficients. Chen [26] constructed two pollutant models to calculate the adjusted mortality coefficients of PM10, SO_2_, NO_2_, and PM2.5 in 17 Chinese cities, discovering that the health effect of SO_2_ had no statistical significance in the two-pollutant model. According to Kan, et al. [31], the excess risk of SO_2_ in Hong Kong almost decreased to zero after the adjustment for NO_2_. Guo, et al. [32] revealed the statistical insignificance of the health risk of SO_2_ on respiratory disease in a two-pollutant model. Therefore, we did not consider SO_2_ in the total health effect.

We applied the adjusted excess risk of NO_2_ and PM10 in the two-pollutant model to calculate the total health effect. Few studies have scrutinized the adjusted coefficients of outpatient visits in the multi-pollutant model, and the costs from outpatient treatment and labor loss have tended to be considerably lower than from hospital admissions and premature mortality. Thus, we did not include economic loss from outpatient treatment in the total estimation of monetarized health burden.

## 3. Results

### 3.1. Air Pollution

As shown in Table 2, the annual average concentration of PM10 of the 14 cities increased from 2010 to 2014 and then decreased from 2014 to 2016. Guigang, Guiling and Liuzhou were the cities with the highest concentrations of PM10. The concentration of SO_2_ kept decreasing from 2011 to 2016, and Liuzhou, Yulin were the cities suffering the worst serious SO_2_ pollution. As to NO_2_, the peak value of concentration for the 14 cities appeared in 2013 or 2014, then went decreasing. Guilin, Nanning and Liuzhou were the cities most seriously polluted by NO_2_.

### 3.2. Health Damage

As the result presented in Figure 2, from 2011 to 2016, cases of all-cause mortality, cardiovascular mortality, respiratory mortality, and all-cause hospital admissions in Guangxi ZAR were 35,327 (19,972–50,528), 14,288 (3631–24,776), 6,146 (2828–9406), and 613,141 (485,446–740,105), respectively. The results showed that the pollutant with the greatest adverse health effect was NO_2_. Cases of NO_2_-related premature mortality, cardiovascular mortality, respiratory mortality, all-cause hospital admissions, inpatients for CVD, inpatients for RD, and total outpatients visits were 46,365 (31,158–61,423), 22,172 (12,408–31,675), 8765 (5058–12,366), 736,973 (598,040–875,052), 19,037 (2395–35,239), 38,044 (22,837–53,074), and 22,784,511 (2,320,249–27,110,320) respectively. The primary reason for this finding was that the coefficient of NO_2_ that resulted in health damage was much greater than that for the other pollutants.

Premature mortality, mortality for CVD, mortality for RD, all-cause hospital admissions, inpatients for CVD, inpatients for RD, and all-cause outpatient visits due to PM10 were 28,396 (14,664–42,014), 15,500 (8144–22,437), 5558 (3096–7991), 418,159 (275,591–559,898), 15,722 (−2063–32,617), 47,083 (33,197–66,748), and 16,397,047 (0–32,595,751), respectively. The SO_2_ causing premature mortality, mortality for CVD, mortality for RD, all-cause hospital admissions, inpatients for CVD, inpatients for RD, and all-cause outpatient visits were 24,618 (15,480–33,371), 11,749 (6682–16,772), 5004 (3140–6886), 548,136 (414,829–675,991), 17,091 (5022–29,292), 46,483 (28,332–64,380), and 16,518,440 (792,306–3,2018,916), respectively, shown in Figure 3.

Nanning suffered the most serious health damage from PM10 with 5047 (2608–7462) cases of premature mortality caused by PM10 (Figure 4). The adverse health burden caused by PM10 of Guilin and Liuzhou ranked second and third among the 14 municipal cities, respectively, whose cases of PM10 caused mortality were both nearly 60% of Nanning, Yulin was most affected by SO_2_, while the cases of premature mortality, reached 4197 (2641–5684). Liuzhou, Guilin, Nanning, and Baise ranking second to fifth, where the cases of SO_2_-caused premature mortality were 3136 (1975–4245), 2760 (1735–3743), 2693 (1691–3654), 2691 (1695–3641) respectively.

The city most affected by NO_2_ was Nanning, whose total premature mortality, reached 10,367. This was much more serious than the situation in Guilin, Yulin, and Liuzhou, which ranked second to fourth in effects from NO_2_ among municipal cities in Guangxi, respectively. Fangchenggang city suffered the least health damage from NO_2_. Its adverse effects from NO_2_ were approximately 1/20 that of Nanning. Further details are provided in the Appendix A.

The peak of the health burden caused by PM10 and NO_2_ appeared in 2014, and the most serious conditions due to SO_2_ occurred in 2013. Among the 14 municipal cities, Nanning, Liuzhou, and Yulin were the most affected by the short-term health burden for ambient air pollutants. Fangchenggang and Beihai were the cities impacted the least health by air pollutants. More details are provided in the Appendix A.

### 3.3. Estimation of Economic Loss

The total estimated economic loss from ambient pollutants in Guangxi Province from 2011 to 2016 was USD $40,555 (24,172–57,585) million, including $4028 (1376–6656) million in resource loss for premature mortality and $36,528 (22,797–50,929) million in labor loss combining hospital expenses and hospital admissions. This accounted for 2.86% (1.70–4.06%) of the province’s GDP. Table 3 shows that the proportions of GDP for economic loss from ambient pollutants decreased from 2011 to 2016.

Among the 14 cities, Yulin, Hechi, Laibing, Baise, and Hezhou suffered the greatest economic losses, with the economic loss from air pollutants in terms of GDP proportions reaching 4.59% (2.83–6.39%), 4.56% (2.81–6.38%), 4.39% (2.54–6.30%), 4.16% (2.55–5.82%), and 4.12% (2.22–6.08%), respectively. Fangchenggang and Beihai were the least impacted economically by air pollution; the air pollution-caused economic loss in terms of proportions of GDP was 1.35% (0.65–2.07%) and 1.37% (0.77–1.99%), respectively (Figure 5).

## 4. Discussion

### 4.1. Health Burden Analysis

As mentioned in the result section, Nanning, Yulin and Liuzhou were the cities suffered the most health damage while Beihai, Fangchengang were the cities with the minimum health effects by air pollution. The health impact of air pollution varied greatly in different cities. The level of air pollution is associated with local socio-economic development. According to previous empirical research, human factors, such as population density, the number of vehicles, and GDP, have had the most significant positive effect on the level of air pollution [33]. In a national-level study, Lin and Dai [34] showed that socioeconomic factors, like energy consumption, industrialization, and technological progress, had variable impacts on air quality. Micro-level research in Shanghai confirmed this view, showing that the level of air pollution in Shanghai showed negative correlation with coastal proximity and positive correlation with population, highway, and industrial intensity [35].

In this study, we attempted to explore the influence of socioeconomic factors on Guangxi’s air pollution using correlation analysis, drawing upon provincial statistical yearbooks [36]. By conducting a regression analysis, we set the major socioeconomic indicators of 14 cities from 2011 to 2016 as independent variables and the ratio of premature deaths caused by ambient pollutants as dependent variables. The example data of socioeconomic indicators we selected are shown in Appendix A. We established a regression model and found that the health effects caused by ambient pollutants in Guangxi were as follows: (1) health losses caused by PM10 and SO_2_ were mainly related to the number of motor vehicles, the urban developed area and the gross industrial production, as shown in Table 4; (2) no socioeconomic indicator from the statistical yearbook that was significantly correlated with SO_2_ caused premature mortality. The correlation analysis shows that health problems and the related economic loss caused by air pollution in Guangxi ZAR were also related to urbanization and industrialization, which is consistent with previous studies in other disciplines.

However, the correlation analysis in this study could not completely elucidate the relationship between socioeconomic development and the health burden from air pollution due to the lack of sufficient data and lack of elaborate analysis, and complete process of demonstration. More intensive studies are needed to fully elucidate the factors underlying the health burden due to ambient pollution.

The timeline shows that heath damage caused by PM10 and NO_2_ reached their peak in 2014, whereas SO_2_-related health damage crested in 2013. The Action Plan for the Prevention and Control of Air Pollution of the State Council of China was promulgated in September 2013 [37], when Guangxi was implementing corresponding control measures. Thus, administrative policies may have played a positive role in controlling air pollution in Guangxi, but this requires further research and supplementary demonstration.

Economic loss associated with air pollution peaked in 2014, nearing USD $7754 (4572–11,083) million, but the proportion of total Guangxi GDP decreased from 3.97% (2.35–5.63%) to 1.89% (1.16–2.67%) from 2011 to 2016, respectively, as shown in Table 3. Among the 14 cities, Yulin, Laibing, and Hechi suffered the greatest losses in terms of the proportional economic burden associated with air pollution. In terms of absolute value, Nanning experienced the greatest economic loss. The correlation analysis showed that economic loss was correlated with urban fraction, pollution density, industry’s share of GDP, and residential income (Table 5).

The results showed that air pollution caused 2.86% (1.70–4.06%) monetary loss in GDP from 2011 to 2016. At the same time, public expenditure for medical security, energy conservation, environmental protection, and science and technology in Guangxi ZAR accounted for 1.42%, 0.50%, and 0.25% of the total GDP, respectively. Therefore, to protect public health and sustain economic development, it is important for the local government to form sound environmental and economic policies to reduce air pollution. They must also increase public expenditures for public health, energy conservation, environmental protection, and technological innovation. Human life is invaluable; its value cannot be defined by a price. This study was intended to demonstrate the adverse effects from ambient pollutants in a straightforward manner, to enhance public awareness and understanding of the hazards posed by ambient pollution.

### 4.2. Uncertainty Analysis

In this study, uncertainty inevitably existed in the health burden results because of the subjective nature of selecting exposure-response coefficients and economic estimation methods. The exposure-response coefficient—one of the core elements in this study—was adopted from a previous study on the same topic. The exposure-response coefficients of all-cause mortality, cardiovascular mortality, and respiratory mortality were adopted from Chen [26], who calculated them from health effects in a two-pollutant model based on 17 cities in China. The exposure-response coefficient related to hospital admission was adopted from other previous studies [9,21,22,23], based on the nationwide situation or on similar spheres in Guangxi. It is advisable to use an exposure-response coefficient derived from a study in the same local area, but there were no similar previous studies in Guangxi ZAR that could be found. In this article, the lower safe threshold of air pollution was used, despite previous studies using safe threshold sets according to the national standard. We think this increases the credibility of our health damage results. Since the idea of safe thresholds of air pollutants remains controversial, the lower value was used in recognition of the seriousness of air pollution’s health effects.

Additionally, uncertainty exists in the economic estimation method. The VSL data were derived from research conducted by Wang and He [29] with local CPI data while uncertainty is inevitable in using VSL method. The value of VSL varied in different previous studies, since it is correlated with the local socio-economic variables such as income level and education level, etc. [11,17,30,38]. Due to the lack of systematic VSL evaluation across Guangxi, the VSL value implemented in this article is from research based on the low-income area where is similar to the actual situation of Guangxi. However, the economic levels in different cities are not the same and this factor can affect the accuracy of the monetary value of health burden. Hospital admission fees were obtained from the China Statistical Yearbook of Public Health. These data apply to the nation as a whole and may not accurately reflect Guangxi ZAR, but the national data were used given the lack of local data.

The three pollutants we selected, PM10, SO_2_, NO_2_, to estimate the air pollution have been monitored by environmental protection department since 2010, other pollutants like CO, O_3_ and PM2.5 started to be included in the scope of monitoring pollutant by Guangxi government in 2017. We do not have sufficient data to analyze air pollution beyond the three pollutants.

We estimated short-term health impacts, whereas long-term effects were not considered. This led to lower estimation results than the actual values. Therefore, to evaluate the overall health impact in future studies, long-term exposure should also be considered.

Besides physical health effect, air pollution also significantly influenced mental health, according to the latest research [39,40]. Since the lack of data to value the economic loss caused by mental health damage in Guangxi, this study only considered the physical health effect, which might lead to underestimation of the actual situation.

The uncertainties mentioned above have been encountered in related studies that estimated the health burden from ambient pollutants, and they will not be resolved unless more detailed and local studies are performed. However, since the models and the data were reliable and authentic, our study still provides an adequate overall picture of the health burden caused by air pollution in Guangxi ZAR. The results of this study can be used not only to inform future regulation in the local area but also to guide related research in less-developed areas in China.

## 5. Conclusions

We estimated the adverse health effects from ambient pollutants—including PM10, SO_2_, and NO_2_—in Guangxi ZAR from 2011 to 2016 using a log-linear exposure-response function. We monetarized the related mortality and morbidity effect by using the value of statistical life (VSL) and cost of illness (COI) method respectively. The health damage results showed that NO_2_ caused more mortality and morbidity than PM10 and SO_2_ for the greater exposure–response coefficient and lower threshold concentrations. Among the 14 cities, residents in Nanning, Liuzhou, Guilin, and Yulin suffered the most serious health damage due to urban development, population, and industry structure. Morbidity caused by air pollutants in Guangxi ZAR peaked in 2013 and 2014, when the Action Plan for the Prevention and Control of Air Pollution under the State Council of China was starting to be implemented. With respect to economic loss, Nanning suffered the most in terms of absolute value, but Yulin, Hechi, and Laibing suffered the greatest losses in terms of the proportion of local GDP: 4.59%, 4.56%, and 4.39%, respectively. These were much higher than the average value of 2.89% in Guangxi ZAR as a whole. Compared to public investments in medicine, environmental protection, and technology innovation, economic loss due to air pollution in Guangxi was considered substantial. Despite uncertainties, the results underscore the severity of the health effects caused by air pollution in Guangxi ZAR. Immediate actions should be taken to alleviate air pollution-related health problems; the results of this study can serve as a starting point in that process. These results can help guide the design and implementation of environmental and economic policies to improve air quality in the region.

## Figures and Tables

**Figure 1 ijerph-16-02707-f001:**
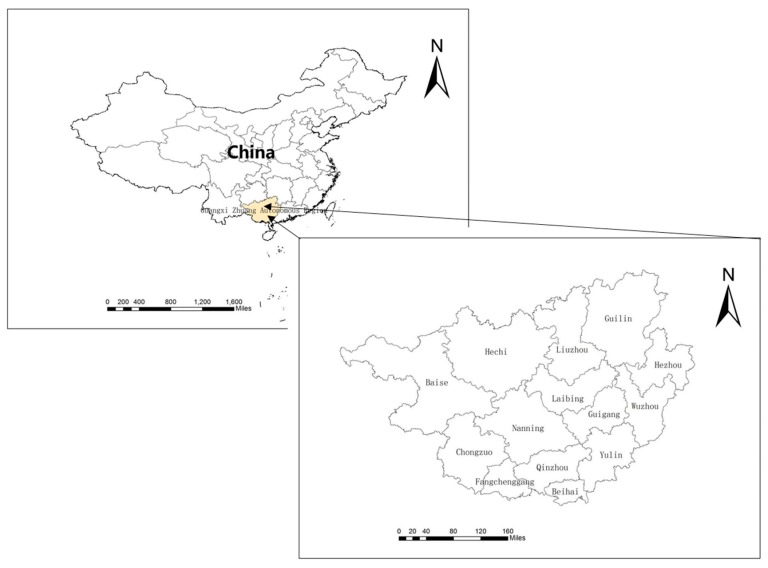
Research study area: Guangxi Zhuang Autonomous Region (ZAR), China.

**Figure 2 ijerph-16-02707-f002:**
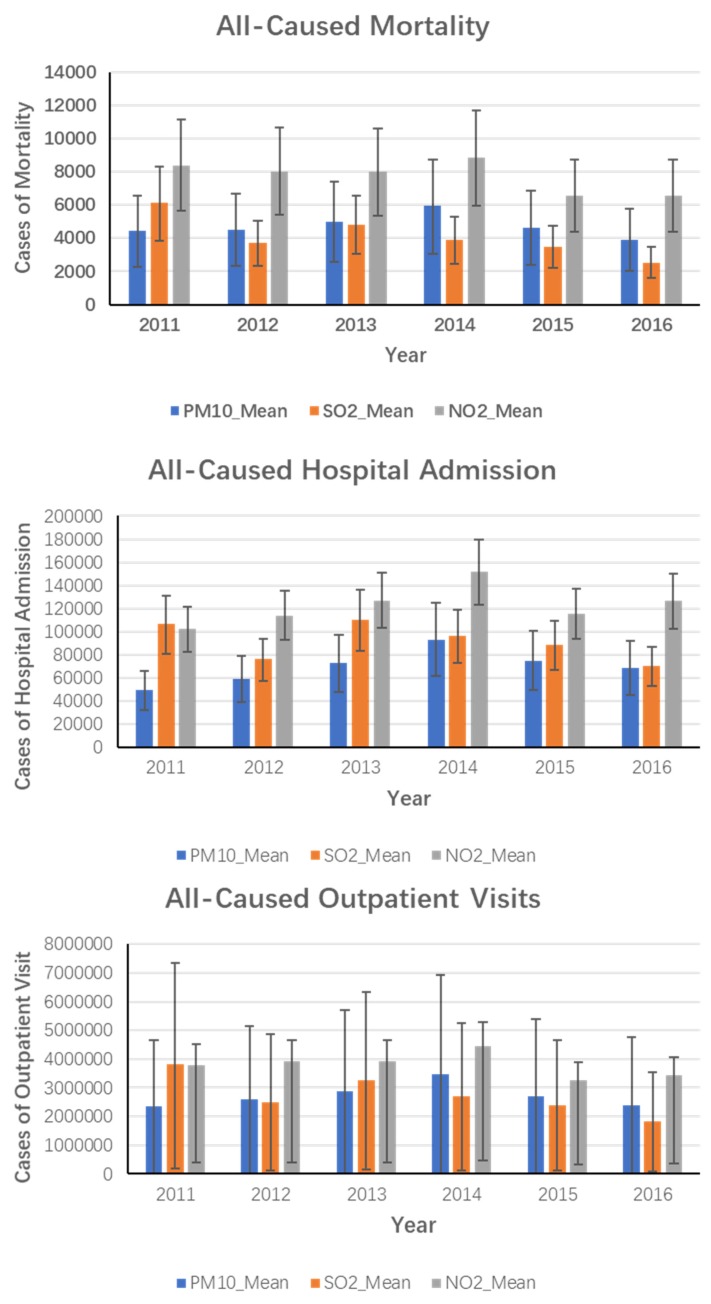
Health effect by air pollutants in Guangxi ZAR from 2011–2016.

**Figure 3 ijerph-16-02707-f003:**
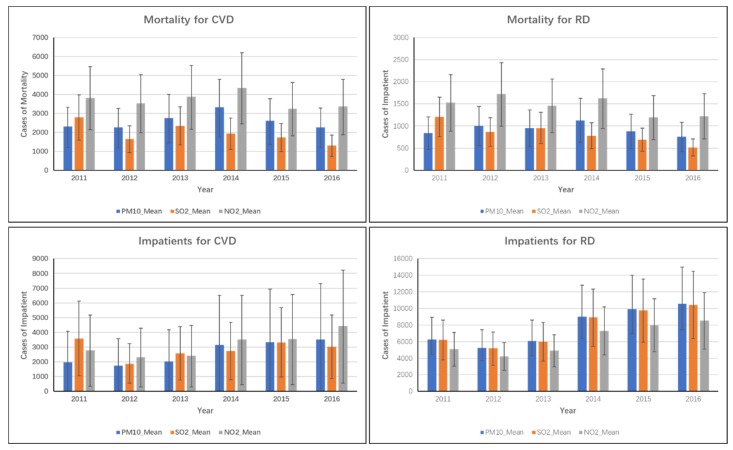
Cardiovascular and respiratory mortality and morbidity caused by air pollutants.

**Figure 4 ijerph-16-02707-f004:**
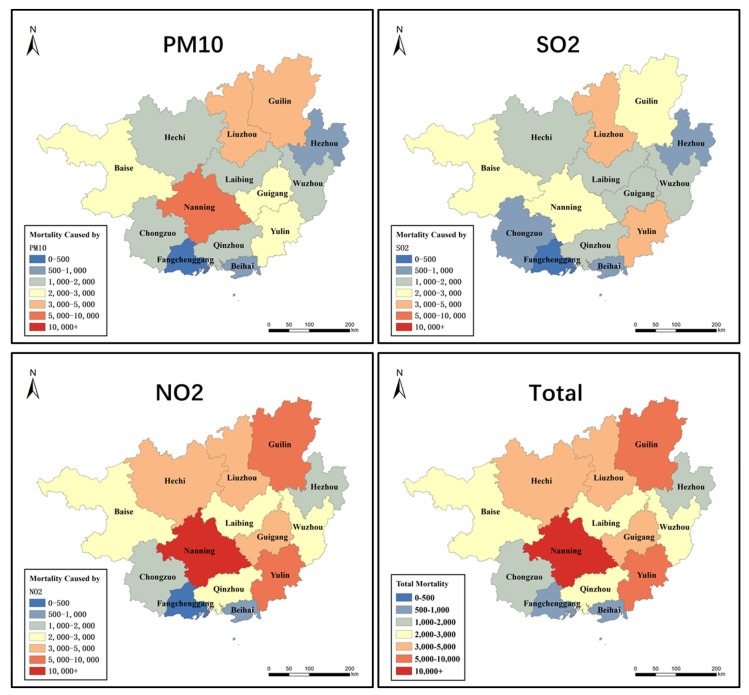
Premature mortality by air pollutants in 14 cities of Guangxi.

**Figure 5 ijerph-16-02707-f005:**
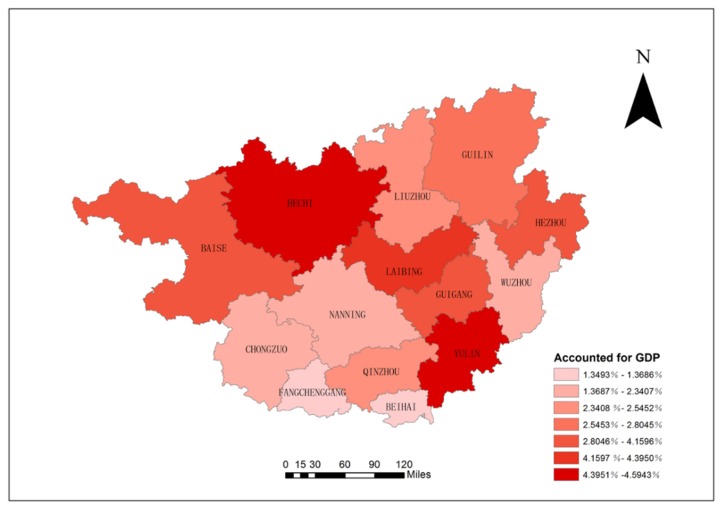
Economic loss (% of GDP) from air-pollution-caused health damage.

**Table 1 ijerph-16-02707-t001:** Coefficients of health effect caused by ambient pollutants.

Health Effect	Pollutant	Coefficient (Per 10 μg/m^3^)	References
All-Cause Mortality	PM10	0.35% (0.18–0.52%)	[26]
	SO_2_	0.75% (0.47–1.02%)	
	NO_2_	1.63% (1.09–2.17%)	
Mortality for Cardiovascular	PM10	0.44% (0.23–0.64%)	
	SO_2_	0.83% (0.47–1.19%)	
	NO_2_	1.80% (1.00–2.59%)	
Mortality for Respiratory	PM10	0.56% (0.31–0.81%)	
	SO_2_	1.25% (0.78–1.73%)	
	NO_2_	2.52% (1.44–3.59%)	
All-Cause Outpatient Visits	PM10	0.25% (0.00–0.50%)	[10]
	SO_2_	0.63% (0.03–1.23%)	
	NO_2_	0.99% (0.10–1.18%)	
All-cause Hospital Admission	PM10	0.35% (0.23–0.47%)	[27]
	SO_2_	1.18% (0.89–1.46%)	
	NO_2_	1.78% (1.44–2.12%)	
Inpatients for Cardiovascular	PM10	0.23% (−0.03–0.47%)	[11]
	SO_2_	0.65% (0.19–1.12%)	
	NO_2_	0.80% (0.10–1.49%)	
Inpatients for Respiratory	PM10	0.77% (0.54–1.10%)	[28]
	SO_2_	0.76% (0.46–1.06%)	
	NO_2_	0.62% (0.37–0.87%)	

**Table 2 ijerph-16-02707-t002:** Annual average concentration of air pollutants of 14 municipal cities in Guangxi (10 μg/m^3^).

Pollutant	Year	2011	2012	2013	2014	2015	2016
PM10	BAISE	59.24	52.00	58.71	92.07	67.53	61.83
	BEIHAI	60.23	52.81	53.27	58.07	48.28	44.42
	CHONGZUO	57.62	54.85	60.48	63.13	55.91	50.76
	FANGCHENGGANG	64.46	54.00	58.30	104.05	49.90	45.15
	GUIGANG	68.26	70.56	78.90	103.22	64.36	55.45
	GUILIN	75.82	70.46	83.65	132.38	66.70	64.12
	HECHI	63.68	63.77	65.37	63.29	68.79	55.24
	HEZHOU	40.63	45.38	51.15	96.29	60.58	54.68
	LAIBING	74.15	55.58	55.79	64.50	60.86	58.12
	LIUZHOU	74.40	71.43	88.02	91.79	69.65	66.47
	NANNING	72.98	68.51	90.08	83.88	71.79	62.18
	QINZHOU	63.61	58.06	62.61	105.07	58.15	54.50
	WUZHOU	42.55	48.04	52.65	68.88	56.63	57.09
	YULIN	51.70	42.01	43.06	97.71	57.09	53.18
SO_2_	BAISE	51.13	13.73	47.28	46.20	16.42	13.10
	BEIHAI	23.11	18.19	17.74	13.47	8.85	9.16
	CHONGZUO	23.59	22.59	14.82	15.39	10.45	11.93
	FANGCHENGGANG	15.04	8.79	9.71	15.00	5.78	9.33
	GUIGANG	18.64	17.09	15.20	16.78	22.00	18.48
	GUILIN	35.23	23.10	27.42	43.33	21.88	17.28
	HECHI	45.08	20.44	26.20	11.75	24.25	12.36
	HEZHOU	16.29	15.09	20.75	19.90	16.30	18.00
	LAIBING	30.46	47.93	25.04	24.11	20.74	21.99
	LIUZHOU	64.01	48.79	34.42	31.35	24.42	21.36
	NANNING	25.76	19.22	19.18	14.89	12.93	12.45
	QINZHOU	19.14	17.88	17.20	33.00	17.36	17.39
	WUZHOU	25.73	14.16	16.24	28.72	17.69	11.03
	YULIN	40.44	43.68	48.68	42.36	32.39	24.83
NO_2_	BAISE	22.15	18.37	20.82	29.35	17.04	14.73
	BEIHAI	19.50	13.21	12.85	14.27	13.95	12.51
	CHONGZUO	18.98	15.20	18.56	19.09	17.70	18.07
	FANGCHENGGANG	23.33	19.17	21.81	21.42	12.32	17.30
	GUIGANG	25.14	17.55	23.97	32.56	20.36	22.19
	GUILIN	30.63	35.64	31.39	50.19	25.15	26.85
	HECHI	27.55	21.12	23.28	23.99	22.45	26.56
	HEZHOU	21.58	20.70	18.98	21.63	14.92	16.46
	LAIBING	32.56	20.85	22.20	24.89	23.96	20.97
	LIUZHOU	32.67	27.37	31.08	30.05	23.73	24.17
	NANNING	32.52	33.15	37.85	37.06	33.05	31.92
	QINZHOU	26.40	23.19	24.56	27.74	19.46	20.05
	WUZHOU	21.62	25.35	25.61	13.39	20.90	21.81
	YULIN	19.22	20.84	21.35	40.71	22.74	23.72

**Table 3 ijerph-16-02707-t003:** Estimated economic loss by ambient pollutant in Guangxi (millions of USD).

Year	All-Cause Mortality	All Caused Hospital Admission	Total	GDP Rate
Total	4028 (1376–6656)	36,528 (22,797–50,929)	40,555 (24,172–57,585)	2.86% (1.70–4.06%)
2011	1096 (457–1730)	5918 (3688–8218)	7014 (4145–9948)	3.97% (2.35–5.63%)
2012	1259 (513–1997)	5599 (3491–7793)	6857 (4004–9790)	3.30% (1.93–4.71%)
2013	555 (117–988)	6890 (4299–9597)	7446 (4416–1,0585)	3.23% (1.91–4.59%)
2014	658 (137–1172)	7096 (4435–9912)	7754 (4572–11,083)	3.03% (1.78–4.32%)
2015	235 (78–392)	6038 (3770–8434)	6273 (3,848–8827)	2.32% (1.42–3.26%)
2016	225 (74–377)	4987 (3114–6974)	5212 (3188–7351)	1.89% (1.16–2.67%)

**Table 4 ijerph-16-02707-t004:** Correlation between health effect by ambient pollutants and socioeconomic indicator.

Pollutant	Socioeconomic Indicator	R
PM10	Gross industrial production	0.69 *
	Developed area	0.63 *
	Number of motor vehicles	0.53 *
NO_2_	Developed area	0.74 **
	Number of motor vehicles	0.74 *
	Gross industrial production	0.65 **

* *p* < 0.05; ** *p* < 0.01 similarly hereinafter.

**Table 5 ijerph-16-02707-t005:** Correlation between economic loss by ambient pollutant and socioeconomic indicator.

Socioeconomic Indicator	*R*
Residential Income	0.86 **
Population Density	0.65 *
Rate of Urbanization	0.60 *
Industry’s share of GDP	0.58 *

* *p* < 0.05; ** *p* < 0.01 similarly hereinafter.

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
