# Peer review of "Estimation of Health Effects and Economic Losses from Ambient Air Pollution in Undeveloped Areas: Evidence from Guangxi, China"

_ijerph, 2019, doi:10.3390/ijerph16152707_

Round 1

Reviewer 1 Report

At this state, it's a draft paper. The methods and the data used are poorly described.

Here are some comments:

Abstract:

1.      You should use the name of the three major pollutants you considered instead of the abbreviations.

2.      You should avoid abbreviations in the abstract (line 20 and 21)

3.      You should explain what the figures in parentheses are (confidence intervals?)

Introduction

1.      Figure 1 and 2 are the same.

2.      Why there is “undeveloped areas” in the title while the description of the Guangxi Zhuang Autonomous Region seems to show the contrary?

3.      Line 45 a reference is needed, and the GDP should be compared regarding other provinces.

4.      Line 51 a reference is needed to support your statement

5.      Line 52, what are the other major cities?

6.      Line 54, you should give the name of the pollutants

7.      Line 56 end of the sentence to be made

8.      Line 57 references are needed.

9.      Line 63 avoid using abbreviations – besides you never define them

10.  You should precise the period of comparison (1975-2005) for 10

11.  Line 87, AHC is not defined

12.  Not so sure about the statement made lines 98-102. See references among others below:

Li, L., Lei, Y., Pan, D., Yu, C., & Si, C. (2016). Economic evaluation of the air pollution effect on public health in China’s 74 cities. SpringerPlus, 5(1), 402.

Wang, Y., Zhang, Y. S., & Li, X. P. (2008). The effect of air pollution on hospital visits for respiratory symptoms in urban areas of Jinan. China environmental science, 28(6), 26.

Materials and methods

1.      Lines 106-107 Why do you only estimate the short-term health effects? Why are the long-term health-effect not considered?

2.      Lines 107-109, you should explain why you use this classification.

3.      Line 126: where is the table S1?

4.      Line 129: reference is missing

5.      Line 132: you should explain what is VSL and what is COI. Why are these methods the most suitable? What are the limits?It not sufficient to write "we use VSL" and that's it. Think about the readers...

6.      Lines 145-146 reference is missing

7.      Lines 147-151 and 151-156 contain many repetitions.

8.      Line 163: you should explain why you focus only on 3 pollutants. Why do you not consider the other pollutants?

Results

1.      Line 205: reference is missing

2.      Line 206: reference is missing

3.      Line 224: reference is missing

4.      You should add a map to locate cities and their main pollutants.

5.      How do you consider neighborhood effects? There is any detail regarding the spatialization of the data.  

6.      Figure 5 is unclear

7.      Line 241: reference is missing

...

The references are to be standardized

I suggest you rewrite the paper taking into account the potential readers of the journal.

Author Response

Dear Reviewer,

       Thanks so much for your constructive advice。 We have made the revision according to your suggestion in the manuscript of new version and supplemental materials. The revision a\ in the files of attachment. Please see the attachment. 

Reviewer#1: At this state, it's a draft paper. The methods and the data used are poorly described.

Here are some comments:

Abstract:

1.           You should use the name of the three major pollutants you considered instead of the abbreviations.

Reply: We have added the full name of the three major pollutants in the abstract.

2.           You should avoid abbreviations in the abstract (line 20 and 21)

Reply: I have made the revision

3.           You should explain what the figures in parentheses are (confidence intervals?)

Reply: We have added the explanation

Introduction

1.           Figure 1 and 2 are the same

Reply:It is the format error, which was disregarded before. We have modified it in this version.

2.           Why there is “undeveloped areas” in the title while the description of the Guangxi Zhuang Autonomous Region seems to show the contrary?

Reply: The GDP per capita of Guangxi was two-third of China’s average level and half of the global average in 2016. It is actual undeveloped area, and we have supplemented the description of Guangxi in Introduction, line 99-105.

3.           Line 45 a reference is needed, and the GDP should be compared regarding other provinces.

Reply: We have added a reference. Considering many international readers have no idea about Chinese provinces, I selected a widely known country to compare here.

4.           Line 51 a reference is needed to support your statement

Reply: We have supplemented the reference.

5.           Line 52, what are the other major cities?

Reply: We have revised the sentence.

6.           Line 54, you should give the name of the pollutants

Reply: We have made the revision.

7.           Line 56 end of the sentence to be made

Reply: We have made the revision

8.           Line 57 references are needed.

Reply: We have made the revision

9.           Line 63 avoid using abbreviations – besides you never define them

Reply: We have made the revision

10.       You should precise the period of comparison (1975-2005) for 10

Reply: We have made the revision

11.       Line 87, AHC is not defined

Reply: We have supplemented the definition of AHC.

12.  Not so sure about the statement made lines 98-102. See references among others below:

Li, L., Lei, Y., Pan, D., Yu, C., & Si, C. (2016). Economic evaluation of the air pollution effect on public health in China’s 74 cities. SpringerPlus, 5(1), 402.

Wang, Y., Zhang, Y. S., & Li, X. P. (2008). The effect of air pollution on hospital visits for respiratory symptoms in urban areas of Jinan. China environmental science, 28(6), 26.

Reply: We have modified the sentence.

Materials and methods

1.                 Lines 106-107 Why do you only estimate the short-term health effects? Why are the long-term health-effect not considered?

Reply: Besides mortality, we also want to study the hospital admission and outpatient visits; Currently, most of the long-term cohort studies focus on PM2.5, hence, no suitable βcan be used to calculate the long-term effect caused by SO2. We understand that it is very important to study the long-term effect. As our result presented, the short-term health effect Is serious enough to draw attention to deal with the air pollution. We will continue to study on this issue, including long-term health effect in future.

2.                 Lines 107-109, you should explain why you use this classification.

Reply: We referred to a lot of literature, and found that the health effects were classified into 3 categories: pre-mortality, hospital admission and outpatient visit, According to much previous study on mortality caused by air pollution, cardiovascular diseases and respiratory diseases have been extensively studied, which provided us appropriate coefficients to conduct our research.

3.                 Line 126: where is the table S1?

Reply: We have added table S1in the supplemental materials

4.                 Line 129: reference is missing

Reply: We have supplemented the reference.

5.      Line 132: you should explain what is VSL and what is COI. Why are these methods the most suitable? What are the limits?It not sufficient to write "we use VSL" and that's it. Think about the readers...

ReplyWe have supplemented the description of VSL and COI method in line 140-155

5.                 Lines 145-146 reference is missing

Reply: We have supplemented the reference.

7.      Lines 147-151 and 151-156 contain many repetitions.

Reply: We have modified the error

8.      Line 163: you should explain why you focus only on 3 pollutants. Why do you not consider the other pollutants?

Reply: Other pollutants started to be monitored by government in 2017, thus we don’t have adequate data to analyze their pollution. We have supplemented the reason in the discussion, line 322-325

Results

1.      Line 205: reference is missing

2.      Line 206: reference is missing

3.      Line 224: reference is missing

Reply: All the missing reference were modified.

4.           You should add a map to locate cities and their main pollutants.

Reply: We have added table 2 to presented main pollutant of each city, in line 198.

5.      How do you consider neighborhood effects? There is any detail regarding the spatialization of the data.  

Reply: In the article, we are using the point observation, each city only has one observation point and the concentration for each city is represented by this point observation. Some previous study also used the point observation for the health burden studies (e.g. Huang et al. 2012 listed in my reference list). Anyway, I agree with your comment that it is necessary to consider the spatialization, since concentration heterogeneity exists for each city. Other techniques should be implemented, such as satellite data and the concentration simulated by 3D chemical transport model. But in our study, lack of techniques supporting, it is difficult to consider the spatialization of the data.

6.      Figure 5 is unclear

Reply: We have remove figure 5

7.      Line 241: reference is missing

Reply: We have supplemented the reference.

The references are to be standardized

Reply: We have standardized all the references.

I suggest you rewrite the paper taking into account the potential readers of the journal

Reply: We have made a lot of revision to the article, according to your comment. And I hope this article can conform to the needs of readers.

Reviewer 2 Report

The authors estimated the  possible short-term health effects of ambient  PM10, SO2, and NO2 concentrations  in  Guangxi Zhuang Autonomous Region,  southwest of China, from 2011 to 2016 using a log linear exposure– response function. They also  applied the two-pollutant model to calculate the  health effect. Finally, the authors  monetarized the economic loss,  including loss of labour resource and cost of medical treatment, due to air pollution using the VSLvalue of statistical life and  COI (cost of illness) methods. Overall, the paper is useful and interesting, however,  I  have a few issues that I hope the authors can address before further considerations.

General remark:

1.      It would be more convenient for the readers if the authors have shown the summary statistics for air pollutants and each city  considered in Table (this also applies to socioeconomic indicators). The first subsection of the Results section should be devoted to these issues.

2.      Several statements and interpretations of results are not clear  to presented results in Figures and Tables and need to be justified (Please see specific comments).

3.      Please pay attention to the editing of texts, because they are  repetitions (e.g. 150-159) and errors (e.g. l. 243, 290, etc.)

My specific and technical comments are listed below:

l. 53-54 …”… found that the volumes of  NO2, CO, and hydrocarbons in some districts of Nanning exceeded the national standard by 8 to 15” The authors should clarify what they mean “volumes”

l. 59 “[4] summarized that…” In my opinion, it would be better: Cao et al [4] summarized that …. Please pay attention to similar remarks in the text (e.g. l. 67, 73, etc).

l. 63 “ 65% of total global PM2.5 was from Asia alone.´ The authors should explain what they mean by 65% of total global PM2.5. Please give a reference.

l. 87 – AHC model – Please give the full name

l. 117 – “ per 10 mg/m3 increase..” - Please, make sure that there is not  per 10 µg/m3 ...See also Table 1.

l. 124  - Please give the values of the  adopted lower limits as the natural background concentrations of air pollutants considered.

l. 126 – Table S1  is missing from the supplementary file provided

l.  135 – The authors should clarify the  annual Consumer Price Index, and how they calculated the VSLs

l. 178-187 - the whole paragraph is illegible, whether these values apply to the entire period (how were they  calculated, because it is difficult to get them  from the drawing?) Why was the premature mortality due to NO2 pollution  higher than cases of all-cause mortality?  

l. 200-211 - the whole paragraphs are illegible. Additionally, 13 cities are marked in Figure 4, and in the legend 14 .

Fig. 5 - there is no legend. In my opinion this Figure can be omitted because it did not contribute anything significant to the ongoing discussion.

l. 247”… have had the most significant positive effect on the air pollution ratio”. - Please develop this issue and define the term “air pollution ratio” and “anti-correlation” (l. 256).

l. 258-267 - These studies should be included in the methods section. The results are not presented, but  there is their discussion. Please define “the industrial added value” , “urban development area”, “urban fraction”, etc..  

l. 338 – CV study - no earlier mentioned about this study.

Author Response

Dear Reviewer,

Thanks so much for your constructive advice。 We have made the revision according to your suggestion in the manuscript of new version and supplemental materials. The revision a\ in the files of attachment. Please see the attachment.

The authors estimated the  possible short-term health effects of ambient  PM10, SO2, and NO2 concentrations  in  Guangxi Zhuang Autonomous Region,  southwest of China, from 2011 to 2016 using a log linear exposure– response function. They also  applied the two-pollutant model to calculate the  health effect. Finally, the authors  monetarized the economic loss,  including loss of labour resource and cost of medical treatment, due to air pollution using the VSLvalue of statistical life and  COI (cost of illness) methods. Overall, the paper is useful and interesting, however,  I  have a few issues that I hope the authors can address before further considerations.

General remark:

1.       It would be more convenient for the readers if the authors have shown the summary statistics for air pollutants and each city  considered in Table (this also applies to socioeconomic indicators). The first subsection of the Results section should be devoted to these issues.

Reply: We have supplemented table 2 to present the air pollution of each city in the Results section.

2.      Several statements and interpretations of results are not clear  to presented results in Figures and Tables and need to be justified (Please see specific comments).

3.      Please pay attention to the editing of texts, because they are  repetitions (e.g. 150-159) and errors (e.g. l. 243, 290, etc.)

Reply: We have made the revision.

My specific and technical comments are listed below:

l. 53-54 …”… found that the volumes of  NO2, CO, and hydrocarbons in some districts of Nanning exceeded the national standard by 8 to 15” The authors should clarify what they mean “volumes”

Reply: We have made the revision to this sentence. The literature we referred here was written in Chinese, we made the translated error here.

l. 59 “[4] summarized that…” In my opinion, it would be better: Cao et al [4] summarized that …. Please pay attention to similar remarks in the text (e.g. l. 67, 73, etc).

Reply: We have made the revision.

l. 63 “ 65% of total global PM2.5 was from Asia alone.´ The authors should explain what they mean by 65% of total global PM2.5. Please give a reference.

Reply: We have added the reference.

l. 87 – AHC model – Please give the full name

Reply: We have supplemented the full name of AHC

l. 117 – “ per 10 mg/m3 increase..” - Please, make sure that there is not  per 10 µg/m3 ...See also Table 1

Reply: We have revised it.

l. 124  - Please give the values of the  adopted lower limits as the natural background concentrations of air pollutants considered.

Reply: We have presented the value of low limits of the three pollutants in supplemental materials, Table S4

l. 126 – Table S1  is missing from the supplementary file provided

Reply: We have added Table S1 in the new edited supplemental document

l.  135 – The authors should clarify the  annual Consumer Price Index, and how they calculated the VSLs

Reply We have supplemented the annual CPI in supplemental materials in Table S3. The calculation of VSLs was clarify in the literature “Wang, H.; He, J. The value of statistical life : a contingent investigation in China; 2010.” referred in line 146.

l. 178-187 - the whole paragraph is illegible, whether these values apply to the entire period (how were they  calculated, because it is difficult to get them  from the drawing?) Why was the premature mortality due to NO2 pollution  higher than cases of all-cause mortality?  

ReplyWe have rewrote the paragraph.

l. 200-211 - the whole paragraphs are illegible. Additionally, 13 cities are marked in Figure 4, and in the legend 14 .

Reply We have revised the paragraphs. Actually, 14 cities are marked in Fig 4, the green marked city, Fangchenggang, whose value is quite small, is easily overlooked. Besides, we have replaced the figure 4 by four maps.

Fig. 5 - there is no legend. In my opinion this Figure can be omitted because it did not contribute anything significant to the ongoing discussion.

Reply: We have remove figure 5

l. 247”… have had the most significant positive effect on the air pollution ratio”. - Please develop this issue and define the term “air pollution ratio” and “anti-correlation” (l. 256).

ReplyWe have rewrote the paragraph.

l. 258-267 - These studies should be included in the methods section. The results are not presented, but  there is their discussion. Please define “the industrial added value” , “urban development area”, “urban fraction”, etc..  

Reply: we have changed the words to “gross industrial production”, “urban developed area” and “rate of urbanization”, which are easier to be literally understood.

l. 338 – CV study - no earlier mentioned about this study.

Reply: we have revised the sentence

Reviewer 3 Report

The study is very interesting, well designed and well written. Some aspects need clarifications:

In Introdution section, lines 46 to 53, authors could provide detailed figures of the pollutants' concentrations in the period of the study.

In the Methods section, lines 133 on, authors could consider other VSL values for comparison purposes, such as the ones published by OECD countries.

In Results section, authors should provide a table with descriptive statistics of the pollutants' concentrations in the analyzed period.

In Discussion section, authors should discuss how differences in VSL could impact on estimates in terms of suggesting public policies for health protection. Adittionally authors should discuss the impact of present levels of air pollution on cities' GDP.

Author Response

Dear Reviewer,

Thanks so much for your constructive advice。 We have made the revision according to your suggestion in the manuscript of new version and supplemental materials. The revision a\ in the files of attachment. Please see the attachment.

The study is very interesting, well designed and well written. Some aspects need clarifications:

In Introdution section, lines 46 to 53, authors could provide detailed figures of the pollutants' concentrations in the period of the study.

ReplyWe have added a table (Table 2) in the Results section to present the concentration of pollutants of each city in the study period.

In the Methods section, lines 133 on, authors could consider other VSL values for comparison purposes, such as the ones published by OECD countries.

Reply: Actually we considered applying the VSL values published by OECD, but the values area much larger than the one we referred. The VSL value we applied, cited from the literature “Wang, H.; He, J. The value of statistical life : a contingent investigation in China; 2010.”, whose survey was conducted in different provinces in China, especially the low-income area. In the undeveloped area of China, some residents have little awareness of health problem and have low willingness to pay for avoiding health risk. Therefore, we think the VSL value we refer is more in line with the actual situation of the area we studied. And we think the value of OECD is more suitable for the research on the whole country and the developed area. Beside, we have added the explanation in line 145-150.

In Results section, authors should provide a table with descriptive statistics of the pollutants' concentrations in the analyzed period.

Reply: We have supplemented the table, in line 200.

In Discussion section, authors should discuss how differences in VSL could impact on estimates in terms of suggesting public policies for health protection. Adittionally authors should discuss the impact of present levels of air pollution on cities' GDP.

Reply: We have made the supplement in Line 333-340.

Reviewer 4 Report

Dear authors, 

Thank you for your submission! Please find my comments below to improve your manuscript.

1. line 35 "guanxi"? Please carefully check your manuscript.

2. Please provide appropriate references in your publication. Please check each reference in the next version. For instance, reference 1 and 2 did not mention the information in line 46-53. 

3. Additionally, could you please also provide the specific information and statistics of the pollution in Guangxi. After all, what you mainly talked about here is about Guangxi, rather than China or the whole world.

4. Line 54-63. Health is not merely about physical health, but also mental and social wellbeing. Could you please also provide empirical evidence of other aspects of health? 

5. Line 46-53. Please also provide information about pollution in Guangxi, particular in  SO2, NO2, PM10 because they are the main indicators of your model.

6. Line 74-93. Here, you mainly talked about the adverse effects of air pollution on physical health. Please carefully organize the structure of your introduction part. First, you want to investigate the Health Effects and Economic Losses from Ambient Air Pollution in Undeveloped Areas. However, what you only talked about is physical health. This is not consistent with your research question. Then,  Undeveloped Areas is your target in this manuscript. However, you did not convince me why you want to conduct this study in this area and why to choose Guangxi. What you talked about in your introduction part is air pollution in China. So, pollution, air pollution, China, Guangxi,  Undeveloped Areas, which one is your real sample? What is your own contribution? I am so sorry that I cannot see any of your own contribution here. Please RE-WRITE your introduction part.

7. Line 105 [14]; [17]. Have you ever checked each detail of your manuscript before your submission? I do not think your academic attitude is appropriate!

8. In your method section, you mentioned: "The exposure–response (E–R) function, which is widely used for health damage assessment, has been applied to calculate the health effects of ambient pollutant". However, I have checked reference 14 and 17, and cannot find out what you have mentioned here. In addition, health and its adverse effects cannot be simply measured in this way because you only consider a part of health and its effects.

9. Line 114 references.

10. Line 115, where is your literature review? 

11. Line 118-119, if you cannot find the information you need, how could you conduct this study?

12. Please RE-CONSIDER your method and RE-WRITE your method section. 

13. I cannot agree with your results and discussion based on this evidence. 

Author Response

Dear Reviewer,

Thanks so much for your constructive advice。 We have made the revision according to your suggestion in the manuscript of new version and supplemental materials. The revision a\ in the files of attachment. Please see the attachment.

Thank you for your submission! Please find my comments below to improve your manuscript.

1.     line 35 "guanxi"? Please carefully check your manuscript.

Reply: Sorry for the mistake, we have made the revision.

2.     Please provide appropriate references in your publication. Please check each reference in the next version. For instance, reference 1 and 2 did not mention the information in line 46-53. 

Reply: we have revised the sentence and reference.

3.     Additionally, could you please also provide the specific information and statistics of the pollution in Guangxi. After all, what you mainly talked about here is about Guangxi, rather than China or the whole world.

Reply:We have made the supplement in line 48-55 and line 100-108.  We select Guangxi to study for it is one of the under developed area in China, and it is near Pear River Delta, most developed area in China. The two regions have similar climate and geographical environment, but their economic development varied enormously. Few previous study focused on under developed area in China. Thus, we select Guangxi as a case.

4.     Line 54-63. Health is not merely about physical health, but also mental and social wellbeing. Could you please also provide empirical evidence of other aspects of health? 

Reply Actuallywe have check some literature that discovered air pollution significantly influenced mental health. We can’t collect sufficient data to value the economic loss of mental health effect, thus we only consider the physical health, while the severity has been presented. We have made the explanation in Discussion section, line 347-350.

5.     Line 46-53. Please also provide information about pollution in Guangxi, particular in  SO2, NO2, PM10 because they are the main indicators of your model.

Reply: We have provide the information in Result section, Table 2, line 202

6.     Line 74-93. Here, you mainly talked about the adverse effects of air pollution on physical health. Please carefully organize the structure of your introduction part. First, you want to investigate the Health Effects and Economic Losses from Ambient Air Pollution in Undeveloped Areas. However, what you only talked about is physical health. This is not consistent with your research question. Then,  Undeveloped Areas is your target in this manuscript. However, you did not convince me why you want to conduct this study in this area and why to choose Guangxi. What you talked about in your introduction part is air pollution in China. So, pollution, air pollution, China, Guangxi,  Undeveloped Areas, which one is your real sample? What is your own contribution? I am so sorry that I cannot see any of your own contribution here. Please RE-WRITE your introduction part.

Reply: We selected Guangxi to study for that no air pollution-health impact study has ever been carried out in their  In Guangxi, the air pollution situation is not less serious than that in developed region, such as Pearl River Delta and Yangzi River Delta , and Guangxi was under pressure both from economic development and environmental protection.

7.     Line 105 [14]; [17]. Have you ever checked each detail of your manuscript before your submission? I do not think your academic attitude is appropriate!

Reply: Sorry for the mistake here, we have checked and modified it.

8.     In your method section, you mentioned: "The exposure–response (E–R) function, which is widely used for health damage assessment, has been applied to calculate the health effects of ambient pollutant". However, I have checked reference 14 and 17, and cannot find out what you have mentioned here. In addition, health and its adverse effects cannot be simply measured in this way because you only consider a part of health and its effects.

Reply: We have revised the reference.

9.     Line 114 references.

Reply: We have added the reference.

10. Line 115, where is your literature review?

Reply: We have revised the sentence.

11. Line 118-119, if you cannot find the information you need, how could you conduct this study?

Reply: Since no previous study focus on metrological relationship between air pollution and health burden in Guangxi can be found. We conducted this study by referring to the coefficients concluded from the studies in China. The method has been applied in several previous literature in my reference list such asHuang, D.; Xu, J.; Zhang, S. Valuing the health risks of particulate air pollution in the Pearl River Delta, China. Environmental Science & Policy 2012, 15, 38-47, doi:10.1016/j.envsci.2011.09.007”, “Lu, X.; Yao, T.; Fung, J.C.H.; Lin, C. Estimation of health and economic costs of air pollution over the Pearl River Delta region in China. Sci Total Environ 2016, 566-567, 134-143, doi:10.1016/j.scitotenv.2016.05.060.”

12. Please RE-CONSIDER your method and RE-WRITE your method section. 

Reply: The method we applied has been implemented by previous studies, and has be has been proved feasible. We have made some revision in the method section and modified the errors.

13. I cannot agree with your results and discussion based on this evidence. 

Reply: We have revised the Result section and provide more data in the manuscript and the supplemental materials

Round 2

Reviewer 1 Report

Thank you for these complete and clear response.

Author Response

Thank you very much for your valuable comments to us.

Reviewer 4 Report

Dear authors,

Thank you for your revision! However, after I looked through your revised version, I did not see your appropriate academic attitudes. Too many mistakes in this manuscript made me really disappointed. (e.g., in your response letter: Dear Reviewer, Thanks so much for your constructive advice。; in your manuscript: line 53, line 115, line 119, line 172....)

In addition, your introduction still did not show the research gaps in your research field. Just leave the relative data and references there, rather than analysis.

Third, you did not provide necessary information about why you want to measure health using mortality of different diseases. This limits the robustness and generalization of your conclusions.

Subsequently, I did not see your contributions in this revised version. You cannot conduct your research just because no people investigated it! You have to point it out your own contributions or how to close any research gaps.

Finally, in table 2, the Annual average concentration of air pollutants of 14 municipal cities in Guangxi increased in 2016. Why will this happen? Have you ever considered it?

Based on these above evaluations, your manuscript has not been fully revised. I am so sorry that I cannot accept it in this version.

Author Response

Dear Reviewer

Thanks for your review with much meticulousness and patience. We realized the importants of academic attitude and proofreand the whole manuscript. Actually, we found more than 50 mistakes in spelling and grammar. We apologize for the lack of strictness in writting, and we have revised the mistaks in the final version. 

According to your advises, we made the modification as follow:

"Thank you for your revision! However, after I looked through your revised version, I did not see your appropriate academic attitudes. Too many mistakes in this manuscript made me really disappointed. (e.g., in your response letter: Dear Reviewer, Thanks so much for your constructive advice。; in your manuscript: line 53, line 115, line 119, line 172....)"

Reply: We have check the mamuscript and revised the mistakes.

"In addition, your introduction still did not show the research gaps in your research field. Just leave the relative data and references there, rather than analysis."

Reply: We have supplemented the contribution of this article in line 112-119

"Third, you did not provide necessary information about why you want to measure health using mortality of different diseases. This limits the robustness and generalization of your conclusions."

Reply: We have supplemented the reason in the Materials and Methods section, line 129-133

"Subsequently, I did not see your contributions in this revised version. You cannot conduct your research just because no people investigated it! You have to point it out your own contributions or how to close any research gaps."

Reply: We have supplemented the contribution of this article in line 112-119

Finally, in table 2, the Annual average concentration of air pollutants of 14 municipal cities in Guangxi increased in 2016. Why will this happen? Have you ever considered it?

Reply: Yes, it is the phenomenon that worths considering.  Annual average concntration of air pollution in 14 cities reached the peak value in 2014 and then decreased. Over half of the cities' annual concentraion of SO2 and NO2 increased in 2016, but comparing with the sharply change since 2014, the increasing rate in 2016 was relative small, we can not    distinguish the change as normal volatility or  significant change in statistic by analyzing the change in only one year.